# Kolmogorov Complexity of Coronary Sinus Atrial Electrograms Before Ablation Predicts Termination of Atrial Fibrillation After Pulmonary Vein Isolation

**DOI:** 10.3390/e21100970

**Published:** 2019-10-04

**Authors:** Katarzyna Stępień, Pawel Kuklik, Jan J. Żebrowski, Prashanthan Sanders, Paweł Derejko, Piotr Podziemski

**Affiliations:** 1Faculty of Physics, Warsaw University of Technology, 00-662 Warsaw, Poland; jan.zebrowski@pw.edu.pl; 2Department of Cardiology, Asklepios Clinic St. Georg, 20099 Hamburg, Germany; p.kuklik@asklepios.com; 3Department of Cardiology, Royal Adelaide Hospital, Adelaide 5000, Australia; prash.sanders@adelaide.edu.au; 4Department of Cardiology and Internal Medicine, Medicover Hospital, 02-972 Warsaw, Poland; pderejko@yahoo.com; 5Institute of Cardiology, 04-628 Warsaw, Poland; 6Department of Physiology, Maastricht University, 6200 MD Maastricht, The Netherlands; piotr.podziemski@gmail.com

**Keywords:** atrial fibrillation, catheter ablation, electrogram complexity, Kolmogorov complexity, Shannon entropy, symbolic dynamics

## Abstract

Atrial fibrillation (AF) is related to a very complex local electrical activity reflected in the rich morphology of intracardiac electrograms. The link between electrogram complexity and efficacy of the catheter ablation is unclear. We test the hypothesis that the Kolmogorov complexity of a single atrial bipolar electrogram recorded during AF within the coronary sinus (CS) at the beginning of the catheter ablation may predict AF termination directly after pulmonary vein isolation (PVI). The study population consisted of 26 patients for whom 30 s baseline electrograms were recorded. In all cases PVI was performed. If AF persisted after PVI, ablation was extended beyond PVs. Kolmogorov complexity estimated by Lempel–Ziv complexity and the block decomposition method was calculated and compared with other measures: Shannon entropy, AF cycle length, dominant frequency, regularity, organization index, electrogram fractionation, sample entropy and wave morphology similarity index. A 5 s window length was chosen as optimal in calculations. There was a significant difference in Kolmogorov complexity between patients with AF termination directly after PVI compared to patients undergoing additional ablation (*p* < 0.01). No such difference was seen for remaining complexity parameters. Kolmogorov complexity of CS electrograms measured at baseline before PVI can predict self-termination of AF directly after PVI.

## 1. Introduction

Atrial fibrillation (AF) is the most complex and common sustained arrhythmia [1] and one of the main causes of cardiac-related hospitalization. Mechanisms of AF perpetuation are still not clear [2,3]. Catheter ablation is one of the most frequently performed treatment procedures in patients with AF [4], with the success rate (non-recurrence after 2 years period) up to 90% in paroxysmal AF and up to 64% in persistent AF after multiple procedures [5]. Haïssaguerre [6] et al. indicated that electrical activity in the region of pulmonary veins (PV) may trigger AF, leading to the need for pulmonary vein isolation (PVI), the cornerstone of AF ablation. In patients with AF at the beginning of the ablation, AF termination is regarded as a favorable endpoint of the procedure. However, PVI does not always lead to the termination of AF, and additional ablation is often performed in hope to improve ablation efficacy. This makes the procedure more difficult, increases the risk of complications and its efficacy is still a matter of debate with some studies questioning the need for additional substrate ablation [7,8].

As ablation emerged as one of the main strategies in AF treatment, studies on signals recorded during this procedure are getting more important and methods related to the complexity of the signal are used with increasing frequency. For example, Lankveld et al. showed that AF complexity parameters, calculated on the basis of surface electrocardiogram (ECG), can verify whether the patient is likely to benefit from catheter ablation [9]. Several methods quantifying electrogram complexity were also developed to find the areas with complex atrial electrograms, which are often considered among the suggested targets for additional ablation sites. The most known is the detection of complex fractionated electrical activity (CFAE) by Nademanee et al. [10]. In 2012, Narayan et al. [2] proposed an innovative approach, implying that a rapid firing foci drives the atria, maintaining AF through short re-entrant circuits (rotors and drivers) [2]. Location of these sources did not coincide with CFAE locations [11]. Ablation of these sources has been suggested as a solution capable of terminating AF without PVI. However, a number of later studies did not confirm this hypothesis, resulting in an ongoing discussion of the applicability of this method [12,13]. In the meantime, pulmonary vein isolation (PVI) is still the reference procedure for catheter ablation of AF. In this view, methods predicting the outcome, duration and extent of an ablation procedure are of clinical importance.

In this study, we measure complexity of a single electrogram at the coronary sinus (CS) at the beginning of the electrophysiological procedure. We propose Kolmogorov complexity approximations (Lempel–Ziv complexity and block decomposition method) as measures useful during the ablation procedure and we compare it with other complexity parameters used in clinical practice. The aim of this study is to assess whether the Kolmogorov complexity of an electrogram recorded in CS can be used to predict termination of AF directly after PVI and to identify the most powerful complexity measure in this regard.

## 2. Materials and Methods 

### 2.1. Study Population and Catheter Ablation Procedure

All recordings were collected in Royal Adelaide Hospital during catheter ablation of atrial fibrillation. Catheter ablation followed fixed, step-wise protocol providing opportunity to investigate the effect of each ablation stage on AF complexity. The study population consisted of 26 patients with AF (paroxysmal 12 patients; persistent 14 patients) with a mean age of 62 ± 9 years. For 17 patients ablation terminated AF. In paroxysmal patients, AF was induced prior to the ablation. The rest of the group was in AF prior to the procedure. Atrial bipolar electrograms were recorded during the whole ablation procedure. A 10-pole catheter with 2–5–2 mm interelectrode spacing was placed distal in CS with the proximal bipole at the CS ostium. For privacy protection, patient personal information was removed from the database.

Thirty second intracardiac electrograms recorded with the sampling rate 1 kHz were analyzed at baseline, before any ablation was done and after each stage of ablation: Ablation of left pulmonary veins (LPV), right pulmonary veins (RPV), ablation at the LA roof, ablation at the fossa region and ablation of complex fractionated atrial electrograms (CFAE). In patients in whom AF was induced prior to ablation, the electrograms for analysis were taken after 5 min of AF perpetuation. Additional 30 s electrograms recorded 30 s prior to AF termination were analyzed as the last stage. In this study, we present results based only on electrograms recorded at baseline and on information about the ablation process (whether only PVI was performed or ablation at more sites was conducted). Results for electrograms recorded prior to termination are presented in Appendix A.

In the calculations, we tested different window lengths and chose 5 s as optimal. See Section 2.3 statistical analysis and Section 3 results for details. All baseline electrograms were assessed with different measures, which were compared between patients in whom AF terminated directly after PVI and those in whom additional stages of ablation were performed.

### 2.2. Complexity of Electrograms

Complexity assessment of AF intracardiac electrograms has mostly been used in previous studies to determine the target area during an ablation procedure [10,14]. In this study, we used the complexity measure as an estimator of the ablation outcome after PVI. For this purpose, we applied a number of measures for atrial electrograms complexity assessment, including such that have already been applied in clinical practice. All these methods are presented below. All parameters used in the calculations are available in the Appendix A.

#### 2.2.1. Linear Methods

For intracardiac signals analysis linear methods have most commonly been used. Although they omit some significant dependencies between signal components and thus are often insufficient, they are relatively easy to determine, and they can give quick and valuable information about the signal. We proposed a classical time-based linear method (mean AF cycle length), and frequency based linear methods (dominant frequency with additional indices).

AFCL (atrial fibrillation cycle length) was measured as the mean of the intervals between consecutive local activations (the time interval between consecutive activation deflections).

DF (dominant frequency) is the frequency at which the largest peak can be found in power spectral analysis after applying FFT (fast Fourier transform) [15]. For more organized AF, lower DF values are expected [16]. Additional indices based on spectral analysis were used to ensure the reliability of DF estimation [17]:

DF RI (dominant frequency regularity index) is the ratio calculated as the area under the estimation of the biggest peak of the power spectrum (with a ±0.75 Hz band width) divided by the area under the whole power spectral density curve from the minimum to the maximum frequency [17,18].

DF OI (dominant frequency organization index) was calculated similarly to RI, but it took into account the power of DF harmonics [18]. 

#### 2.2.2. Non-Linear Methods

During AF nonlinearity of the heart increases [19]. Based on that, we proposed non-linear methods as being more appropriate in the context of the electrogram complexity analysis. Looking for a measure that can give better insight into the nature of the electrogram, we proposed some information content and entropy methods (Lempel–Ziv complexity, block decomposition method, Shannon entropy and sample entropy), one method based on signal morphology (wave morphology similarity index) and a number of CFAE parameters.

Kolmogorov complexity K(s) of a string s is defined as the length of the shortest binary description of s [20]. It is an absolute and objective quantification of the amount of information in this string [20]. We proposed two methods to approximate Kolmogorov complexity of an electrogram: Lempel–Ziv complexity and block decomposition method. Both methods require input data to be defined on an alphabet. Thus, to enable them to assess electrogram complexity, before calculations, a conversion to 0–1 alphabet (a binary string) was performed. The conversion procedure is presented below.

First, downsampling to frequency 0.5 kHz, i.e., removing every second sample from the raw signal, was performed. Then an electrogram was transformed into a series of “0” and “1” (binary string s), where the 1 s correspond roughly to local activation events, according to the formula:(1)$$s_{i}=\{\begin{array}{c} 1;{\mathrm{ISP}}_{i}>A_{i}\\ 0;{\mathrm{ISP}}_{i}\leq A_{i} \end{array},$$
where A_i_ is a moving threshold, calculated for every signal sample, and ISP is the instantaneous signal power of the electrogram [21]. ISP can be statistically interpreted as a moving variance and has the properties of a high-pass filter. The assumption is that the spikes of the ISP curve reflect the depolarization events [22]. ISP is then compared with an adaptive threshold A_i_, describing the power of background noise for a particular moment in time. To find A_i_, application of an adaptive procedure should be performed. This procedure reduces noise, acting as a high-pass filter [21], according to the rules:(2)$$M_{0}=X_{0};M_{i}=M_{i-1}+D_{1}\left(X_{i}-M_{i-1}\right),$$
(3)$${\mathrm{ISP}}_{0}=0;{\mathrm{ISP}}_{i}={\mathrm{ISP}}_{i-1}+D_{1}\left({\left(X_{i}-M_{i-1}\right)}^{2}-{\mathrm{ISP}}_{i-1}\right).$$

M (adaptive mean) and ISP are defined with D_1_ as an adaptation constant (0 < D_1_ < 1; according to the authors of the method, in our analysis D_1_ = 0.75 was used) [21]. X is the raw signal after initial noise reduction and normalization. The moving threshold is defined as:(4)$$A_{i}={\mathrm{MISP}}_{i}+0,1V_{i},$$
where MISP is the mean instantaneous signal power and *V* is the adaptive power variance [21]. These quantities were calculated using:(5)$${\mathrm{MISP}}_{0}={\mathrm{ISP}}_{0};{\mathrm{MISP}}_{i}={\mathrm{MISP}}_{i-1}+D_{2}\left({\mathrm{ISP}}_{i}-{\mathrm{MISP}}_{i-1}\right).$$
(6)$$V_{0}{}^{2}=0;V_{i}{}^{2}=V_{i-1}{}^{2}+D_{2}\left({\left({\mathrm{ISP}}_{i}-{\mathrm{MISP}}_{i}\right)}^{2}-V_{i-1}^{2}\right).$$

The parameter D_2_ was set at 0.02, according to the authors of the method [21] (the optimal value of D_2_ was found based on several tests performed on selected signals scored by an expert cardiologist). The converted binary signal has the same length as the downsampled raw signal, for instance a 5 s electrogram recorded with the frequency of 1 kHz corresponds to 2500 samples. To approximate K(s) of the binary string, we proposed the following approaches:

LZC (Lempel–Ziv complexity) is the compression algorithm that has traditionally been used to approximate the Kolmogorov complexity of an object [23]. Recent studies [23] have shown a number of limitations of compression algorithms in the light of Kolmogorov complexity, however under specific conditions it is still a very good estimation of K(s). LZC is based on the method introduced by Pitschner and Berkowitsch [21] to automatically quantify the degree of AF organization. The method applies the principles of symbolic dynamics [24], transforming the electrogram into a binary string and calculating the complexity of such a string. 

After conversion, complexity for a binary string is calculated using the Lempel–Ziv algorithm [21,22]. This is done by finding the number of the so-called words in this string. For each element of the string, a check is performed whether the new sequence (composed of the word already analyzed and the next symbol) has appeared in the signal before. If not, then that sequence becomes the new word and complexity is raised by 1. Otherwise, another symbol is added and the check whether the sequence has occurred performed again. The calculation of LZC can be mathematically described by formula:(7)$${\mathrm{LZC}}_{0}=1;{\mathrm{LZC}}_{k+1}=\{\begin{array}{c} {\mathrm{LZC}}_{k};S_{l+1}\ldots \;S_{k+1}\epsilon S_{1}\ldots S_{k}\\ {\mathrm{LZC}}_{k}+1;S_{l+1}\ldots S_{k+1}\notin S_{1}\ldots S_{k} \end{array},$$
where *l* is the index of the last sample of the last word found. For example, the sequence S1 = 11001 with length n = 5 can be split into three different sequences: (1)(10)(01), providing Lempel–Ziv complexity LZC_S1_ = 3. In contrast, the simple sequence S2 = 00000 of the same length should be transformed into the words (0)(0000), which gives the result of LZC_S2_ = 2.

In a comparative analysis of the signals, normalized LZC may be useful. It can be defined as:(8)$${\mathrm{LZC}}_{\mathrm{norm}}=\frac{\mathrm{LZC}}{B},$$
where normalization factor B is the maximal number of words for a symbol sequence in a binary alphabet, and can be calculated as: (9)$$B=\frac{n}{\log_{2}n},$$
where n is the length of the signal [22]. Key steps of the LZC method are depicted in Appendix B in Figure A1.

Figure 1 illustrates the differences between two 2 s electrograms of different complexities (Figure 1A,B), comparing graphically the results of the LZC method (Figure 1C–F). Despite an apparently similar level of complexity assessed during the visual inspection of Figure 1A,B, a quantitative analysis of LZC shows that there is a significant difference between the complexities of those signals.

BDM (Block Decomposition Method) is a modern information theory method, which aims to find a set of small computer programs that produces the components of a larger object [23]. The first studies using BDM for the analysis of physiological signals have been published recently [25]. BDM is an extension of the coding theorem method (CTM), which has first been used for the compression of very short strings. CTM is calculated based on the algorithmic frequency of production of a string s and its algorithmic (Kolmogorov) complexity K(s), which is defined as the length of the shortest program p that outputs the string s on a universal Turing machine U. CTM approximates Kolmogorov complexity using the following dependency: The more frequent the string is, the lower Kolmogorov complexity it has. Formally, CTM for a binary string s is represented by the following formula:(10)$$\mathrm{CTM}\left(s,t,k\right)=-\log_{2}D\left(t,k\right)\left(s\right),$$
where (t, k) is the space of all t-state k-symbol Turing machines, t, k > 1 and D(t, k)(s) is the function assigned to every finite binary string s [23].

Since CTM is computationally expensive, the block decomposition method has been developed to manage reproducing larger objects. BDM decomposes an object into smaller programs, strongly relying on CTM-using it to calculate algorithmic complexity approximations of smaller pieces of a large object and reconstructing an approximation of the Kolmogorov complexity for the larger object. Fixing t and k, we could define the 1-dimensional BDM of a string s with the following formula:(11)$$\mathrm{BDM}\left(s,l,m\right)=\underset{i}{\sum }\left[\mathrm{CTM}\left(s^{i},m,k\right)+\log\left(n_{i}\right)\right],$$
where n_i_ is the multiplicity of s^i^, s^i^ is the subsequence i after decomposition of s into subsequences s^i^, each of length l, and m is an overlapping parameter (m = l means no overlapping) [23]. 

In this study, after converting the electrograms to binary strings, BDM was calculated using software available online and described in [23,26]. We chose l = 1*2* as optimal decomposition length. 

ShEn (Shannon entropy) is a statistical measure of information uncertainty. Shannon entropy quantifies the properties of the probability distribution of the signal (a normalized histogram) providing a measure of information content [27]. Electrograms in which the signal has a few states (i.e., narrow, regular deflections of similar amplitude) have a narrow distribution in the voltage histogram, and low ShEn values. Electrograms in which the signal adopts a broad distribution of states (irregular morphology and amplitude, fractionation) have a wide distribution in the voltage histogram, and large ShEn values [28].

SampEn (sample entropy) requires the selection of three parameters: The length of the sequences to be compared (m), the patterns similarity tolerance (r) and the number of samples under analysis (N), which in our study were chosen according to Alcaraz et al. [29]. Sample entropy is based on the conditional probability that if two sequences of signal samples of length m (taken at two different time points of a signal) are similar (the amplitude of the corresponding samples is not different than by a predefined threshold), these two sequences will remain similar if the sequences of length m *+ 1* will be considered at the same time points [30]. The more organized AF is, the lower is sample entropy [16].

WMSI (wave morphology similarity index) is an algorithm for the evaluation of the organization of atrial electrograms during AF based on the similarity of electrogram morphology. It relates morphologies of all possible pairs of electrogram deflections extracted from the recording and estimates their similarity. The algorithm describes the regularity by measuring the extent of repetitiveness over time of its consecutive activation waves morphology [31].

CFAE (complex fractionated atrial electrograms) have been suggested to indicate tissue areas associated with the AF substrate [10]. CFAE are postulated to represent areas contributing to AF maintenance, however this relationship is still not fully understood and it is questioned [32]. There is also no consensus on the definition of CFAEs. One of the proposed definitions describes CFAEs as the atrial fractionated electrogram composed of two or more deflections and/or a perturbation of the baseline with a continuous deflection of a prolonged activation complex [27]. A second method describes CFAE as an electrogram with a very short cycle length (smaller than 120 ms) [27]. We used the second definition in our study. We investigated five approaches to quantify CFAEs:

NavX CFAE (NavX complex fractionated atrial electrograms)—mean complex fractionated electrogram calculated as the averaged time interval between the marked deflections, given in percent [32], used in the EnSite NavX electroanatomical mapping system. In the figures, the abbreviation “NAVX” is used.

The following four algorithms are accommodated in the CARTO electroanatomical mapping system:

CEA (continuous electrical activity)—defined by the presence of two or more successive deflections for which the interval length is shorter than 50 ms. CEA is expressed as percentage of continuous activity [32].

ICL (interval confidence level)—the number of intervals identified between consecutive complexes defined as CFAE [10],

ACI (average complex interval)—the average of all intervals that have been identified between consecutive CFAE complexes, given in ms [32],

SCI (shortest complex interval)—the shortest interval of all that have been identified between consecutive CFAE complexes, given in ms [10].

### 2.3. Statistical Analysis

The difference between complexity parameters was tested using ANOVA. Patients were divided into two groups: The first group consisted of 16 cases for which AF terminated directly after PVI (nine patients with paroxysmal AF and seven patients with persistent AF) and the second group of 10 patients for whom an isolation of the PV was not sufficient to terminate AF and further steps of ablation were attempted (three patients with paroxysmal AF and seven patients with persistent AF). In all cases, the statistical significance lower than 0.05 was considered as significant.

To check whether the group separation factor (defined as ablation at additional sites) was chosen correctly, sensitivity and specificity was calculated and ROC (receiver operating characteristic) analysis was performed. Accuracy was measured as the area under the ROC curve (AUC). AUC larger than 0.7 signified that the group separation factor was accurate. For those methods for which we found statistical significance or a trend, the optimal point for ROC (OPT ROC) was calculated as the cut-off point closest to the true positive rate of 1 and false positive rate of 0.

All parameters were calculated in 2, 5, 10 and 30 s segments (the first single-window of the signal). Additionally, the results were accompanied by the mean of the moving window segments over the full 30 s recording, with an overlap equal to one half of the window length.

## 3. Results

There was a significantly higher Kolmogorov complexity (approximated by LZC and BDM) of CS electrograms in patients for whom AF did not terminate after PVI alone and who underwent ablation at additional sites than for the patients for whom AF terminated after PVI (*p* < 0.05 for each tested window length; see Figure 2A,B). In case of Shannon entropy, a tendency towards significance was found (e.g., for 5 s window p = 0.052; see Figure 2C). There was no significant difference between groups for all other parameters (AFCL, DF, RI, OI, all CFAE parameters, SampEn and WMSI; see Figure 3).

ROC analysis resulted in an accuracy of 0.763 for LZC, 0.806 for BDM and 0.756 for ShEn (see Figure 4). Results for the other parameters (AFCL, DF, RI, OI, all CFAE parameters, SampEn and WMSI) are shown in Figure 5. For all parameters other than LZC, BDM and ShEn, the area under the ROC was lower than 0.7. 

Table 1 summarizes the results of ANOVA and ROC analysis obtained for all 14 examined parameters for the first 5 s window of the 30 s signals. For LZC, BDM and ShEn, the sensitivity, specificity and the value for OPT ROC are also presented.

To explore how the length of the analyzed electrogram affects the performance of each tested method, we compared the values of AUC for every method for four different signal lengths: 2 s, 5 s, 10 s and 30 s (Figure 6). The same check was performed for ANOVA (*p*-value) results (a graphical presentation is available in Figure A2 in Appendix C). For both approaches, in case of LZC, BDM and ShEn the differences between the results obtained for particular window lengths were insignificant. Correlation analysis revealed that both methods that approximate Kolmogorov complexity, i.e., Lempel–Ziv complexity and the block decomposition method, were very strongly correlated (the correlation coefficient equaled 0.98 or more for all examined window lengths). A strong correlation between those methods and Shannon Entropy was also found (with the correlation coefficient larger than 0.71 for both examined pairs for different window lengths). 

The tables with the results for ANOVA and ROC for all windows are presented in Appendix D (Table A1) and the correlation tables for all methods are available in Appendix E (Table A3, Table A4, Table A5 and Table A6). In addition, for each method the mean values of the results obtained using a sliding window for the same window lengths were calculated. Results can be found in Table A2 in Appendix D. The approaches using the first window and a sliding window gave comparable results. In this study, we decided to focus only on the first 5 s of the signal, as we perceived this time period as optimal both to quickly obtain complexity parameters and to contain a significant number of signal activations.

## 4. Discussion

Understanding the mechanisms initiating and sustaining AF remains a challenge. Despite over 100 years of research in this field, there is still disagreement on the basic AF mechanisms [2,3].

Catheter ablation of AF, one of the main AF treatment strategies, is a complex and difficult procedure. Its favorable end-point is an abrupt termination of AF, which has been linked with long-term success [33]. However, several studies demonstrated that extensive ablation does not improve long-term efficacy [8,33].

As the number of various approaches to AF catheter ablation increases [2,5,10,34,35,36], methods predicting procedure success are valuable in the context of patient selection and procedure planning. Recent studies often focused on predicting success during long-term follow-up [37,38]. In our study we do not indicate a long-term success of the procedure but that the ablation procedure duration would be extended beyond pulmonary veins. Specifically, we aimed to predict, using only electrogram properties measured at a single site, if the sole ablation of PVI will lead to AF termination directly after pulmonary vein isolation. Due to unsatisfactory atrial fibrillation ablation results, especially in patients with persistent AF, currently in the literature the necessity of an individualized approach to AF ablation is emphasized. Hence, our study is a step in this direction [39,40].

In our study, we investigated whether various complexity parameters could predict catheter ablation outcome. Receiver operating characteristic analysis showed that Kolmogorov complexity approximations and Shannon entropy distinguished reasonably well patients in whom PVI terminate AF from those in whom it was not sufficient for AF termination. For those methods, the area under ROC was larger than 0.75. However, ANOVA analysis demonstrated that a significant difference between such patients, for whom AF spontaneously terminated directly after PVI, in comparison to those in whom additional steps of ablation were undertaken, can be found only for Kolmogorov complexity (*p* < 0.011). The remaining complexity parameters examined did not differ between the groups, with Shannon entropy showing a result marginally close to significance (*p* < 0.053). Between Kolmogorov complexity and Shannon entropy a strong correlation was found (with a correlation coefficient oscillating about 0.8).

### 4.1. Kolmogorov Complexity in AF Electrogram Complexity Assessment

In our study, we demonstrated that Kolmogorov complexity of electrograms recorded at the beginning of the procedure could suggest how a patient will respond to PVI and a more extensive ablation. We presented two approaches of approximating Kolmogorov complexity K(s) of a binary string s. The first of them is a lossless compression algorithm that has traditionally been used to approximate K(s), providing its upper bounds. However, this algorithm has some limitations, specifically it fails to estimate K(s) of small objects and has been proven to be closer to entropy estimators than to K(s) recently [23]. The second proposed K(s) estimation is the block decomposition method, a measure that combines Shannon entropy in the long range with local estimations of algorithmic complexity and, contrary to compression algorithms, can deal with any signal length [23]. In this study, the binary signals we operated on were relatively long (1000 to 15,000 samples), which resulted in very strong correlation between LZC and BDM. Therefore, in the context of the database examined, we considered LZC to be an as good approximation of Kolmogorov complexity as BDM, and we left the open question which algorithm will be better in clinical applications: LZC, which is conceptually and computationally easier, or BDM, more complex, but a universal method. However, we wanted to highlight again that LZC and BDM are not equivalent in most cases, especially for very short signals.

In this study, we proved that Kolmogorov complexity of AF was strongly related to the complexity of the signal in a clinical context. We compared this measure with other methods applied to the analysis of electrogram complexity.

### 4.2. Literature Overview

The set of 12 other methods chosen in this paper to be compared with Kolmogorov complexity represents different approaches based on time and frequency analysis (AFCL, DF, RI and OI), wave morphology and recurrence (WMI), entropy (ShEn and SampEn) and a number of CFAE measures as defined by the NavX or CARTO systems.

Due to the suboptimal efficacy of AF catheter ablation, much effort goes into studies attempting to identify the regions of atria responsible for AF maintenance as promising targets for ablation. The most known method of identifying such regions is complex fractionated atrial electrograms (CFAE), a measure reported by Nademanee [36]. It generated a very widespread response, and extensive studies were performed to verify whether CFAEs indicate AF-perpetuating sites. However, further studies did not reproduce the success, with a recent randomized trial (STAR AF II) demonstrating no extra benefits of additional CFAE ablation over pulmonary vein isolation alone in persistent AF patients [8]. Narayan et al. showed using monophasic action potential (MAP) catheters that sites showing CFAE reflect rather far-field signals, AF acceleration or disorganization than localized rapid AF sources [41]. Lau et al. showed that CFAE correlates poorly with substrate complexity measures like conduction velocity or electrical dissociation [32].

Common use of CFAE methods proves that the clinicians consider signal complexity as an important diagnostic value. However, definitions of CFAE measures are determined mostly on the basis of clinical practice. This suggests that the use of other methods that are able to measure complexity in the context of electrogram analysis is of outstanding importance.

There have been a number of studies that focused mainly on the regularity of AF [42,43,44]. Another popular group of methods used in AF electrogram complexity assessment are entropy-based measures. Several entropy-based methods have been defined in the literature, such as: Approximate entropy (ApEn), Shannon entropy (SE), sample entropy (SampEn) and multiscale entropy (MSE). Although all of them are related to the same concept, the mathematical formulations vary among them [45]. In the case of AF electrograms, entropy may be associated with the disorganization of the atrial electrical activity [45]. An entropy-based approach in the electrogram assessment was represented, for example by Cervigón et al., who proposed sample entropy measured in the right atrium as a predictor of AF recurrence outcome after PVI [37]. In another study, Ng et al. reported ShEn to be able to identify CFAE sites for ablation automatically [27]. A study by Ganesan et al. showed that the pivot of the rotor is consistently associated with high Shannon entropy of bipolar electrograms [46]. In another study concerning rotors localization, approximate entropy has been proposed [47]. An improved version of multiscale entropy was used in a study aimed to discriminate fractionated electrograms in paroxysmal versus persistent atrial fibrillation [48].

In the meantime, linear methods are still often used in AF complexity analysis. For example, Matsuo et al. [49] reported cycle length measured on the surface ECG to be a predictor of long-term success in persistent AF patients. As AF is associated with electrical remodeling, which is reflected in significant signal information during spectral analysis of AF, in many recent studies spectral indices (like DF, RI and OI) have been used [9,38,50]. What is important, Szilágyi et al. showed that spectral measures performed better than clinical factors for predicting AF recurrence.

### 4.3. Physiological Meaning of the Results

One of the main aspects of the physiological meaning of our results is a link between electrogram complexity and likelihood of termination after PVI. Several studies demonstrated a link between the complexity of electrogram morphology and the degree of electroanatomical remodeling of the underlying myocardial tissue. This includes tissue fibrosis, endo-epicardial dissociation, altered action potential kinetics and repolarization abnormalities. All these factors lead to the formation of a substrate for extra-pulmonary AF drivers. Thus, the higher the complexity of CS electrograms, the higher likelihood of extra-pulmonary veins drivers and thus the lower the chance of AF termination after PVI.

The understanding of how much AF is dependent on triggers localized in the region of PVI is limited by diverse ablation methodologies that do not seem to result in durable pulmonary vein isolation [51]. In this view, it is important to search for indicators of other mechanisms of AF perpetuation than those localized in PVI. Our study indicates, that an immediate success of the termination of AF due to PVI isolation may be predicted by a remote measurement in the CS, and therefore the existence of other mechanisms for AF perpetuation than these limited to the PVI region can also be detected remotely by the analysis of the complexity of the electrogram. If this hypothesis is true, more complex mechanisms of AF perpetuation, whether localized or not, will result in a non-local increase of electrogram complexity, and mechanisms that perpetuate AF affect the conduction on a global, atrial level, rather than only locally either functionally or mechanistically.

### 4.4. Clinical Implications

The required time length of the measurement is one of the key elements of practical usability for algorithms analyzing electrograms in a clinical context. Therefore, we performed calculations not only on the whole 30 s recordings gathered during each ablation procedure, but also on shorter portions of the electrogram. We analyzed the first 2 s, to check if such a short signal (relevant in case of e.g., problems with maintaining a good contact between the catheter and the atrial wall) can be useful, as well as 10 s and 30 s signals to see if there is an increased value in analyzing longer recordings. The results for all examined windows were comparable, so we chose the 5 s length as optimal. In further study, it may be interesting to check how Kolmogorov complexity changes during the whole measurement. The area under the ROC for all window lengths for each parameter are indicators of the stability of the method and its vulnerability to the length of measurement.

Our study indicates that Kolmogorov complexity of an electrogram measured at CS at baseline was larger when AF did not spontaneously cardiovert after PVI (see Figure 2, Table 1). As a result, this indicates that additional steps of ablation after PVI might be required to terminate AF. Thus, this indirectly indicates that the required ablation procedure might take a longer time. Therefore, the measurement of Kolmogorov complexity could be used as an indicator of the expected ablation duration and how demanding the procedure might be. Based on the optimal ROC point (OPT ROC point) calculation, we found LZC equal to 75 and BDM equal to 2076 as optimal values on the ROC when using the first 5 s window of the signal.

The results of this study represent a step towards standardization of AF electrogram analysis, needed to adequately address the clinical relevance of ablation performance assessment and management [16].

## 5. Conclusions

Low Kolmogorov complexity of an electrogram measured at the coronary sinus at the beginning of ablation can predict whether a non-induced termination of AF will occur directly after pulmonary vein isolation. The result obtained, combined with the short recording time required for Kolmogorov complexity calculation (5 s) and the fixed measurement site (coronary sinus) made this method potentially applicable in clinical practice.

## Figures and Tables

**Figure 1 entropy-21-00970-f001:**
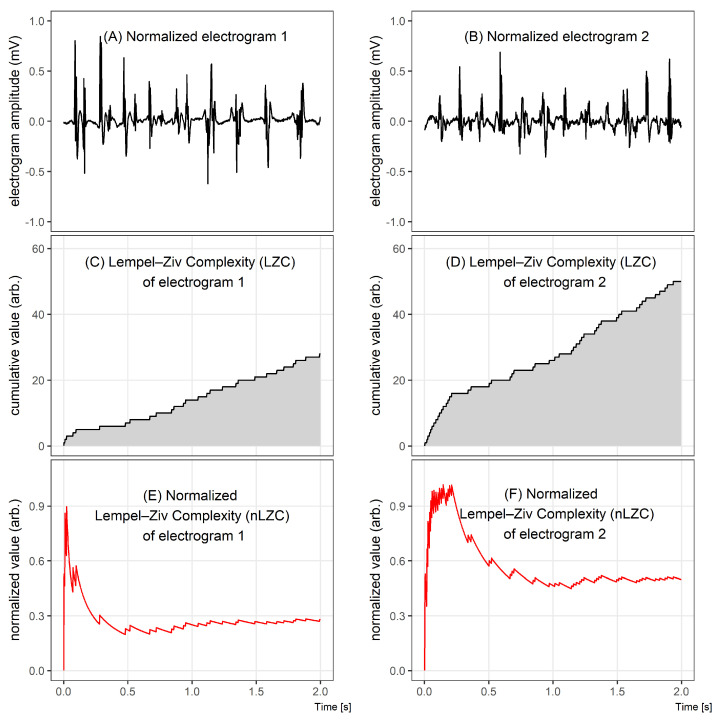
Comparison of two electrograms with different Lempel–Ziv complexities. In the top graphs, 2 s windows of bipolar electrograms are shown. In the middle, the corresponding Lempel–Ziv complexity (LZC) plots for both cases and, in the bottom figures, the normalized LZ complexity plots. The normalized LZC at the beginning of the given fragment rapidly increased in both signals, but, for signal 1, it next significantly decreased, while, for signal 2, it oscillated around half of the maximum value.

**Figure 2 entropy-21-00970-f002:**
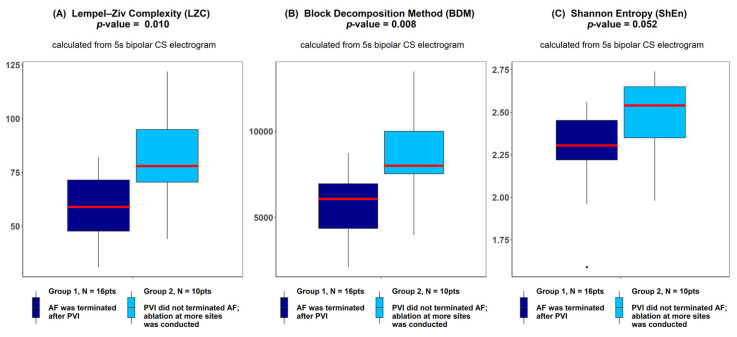
Lempel–Ziv complexity (**A**), block decomposition method (**B**) and Shannon entropy (**C**) of 5 s coronary sinus electrograms for patients for whom atrial fibrillation (AF) terminated after pulmonary vein isolation (PVI) and for patients for whom ablation at more sites was conducted. ANOVA analysis revealed a significant difference between groups in LZC and the block decomposition method (BDM) and a trend towards significance was found for the Shannon entropy (ShEn) method. The thick red line represents the median for each group, outliers are marked as individual small circles; box boundaries correspond to Q1 and Q3 (lower and upper quartile); the upper whisker is located at the smaller of the maximum group value and Q3 + 1.5 IQR (interquartile range) value, whereas the lower whisker is located at the larger of the smallest group value and Q1 -1.5 IQR. Q1 and Q3 are the first and the third quantile, IQR is the length of the box and equals Q3-Q1.

**Figure 3 entropy-21-00970-f003:**
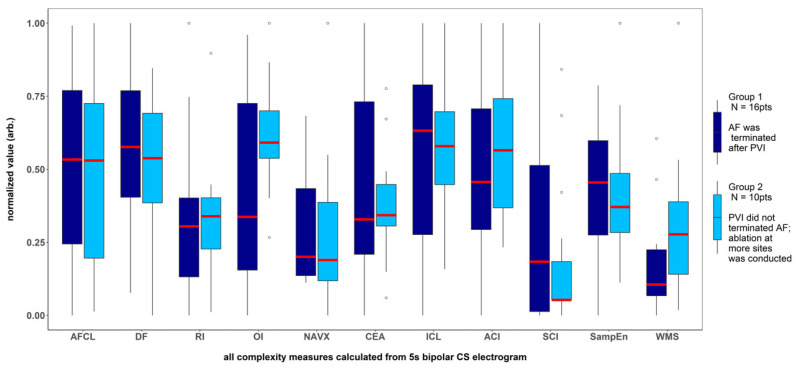
Results for the parameters for which no significant difference was found between the patients with AF terminated after pulmonary veins isolation and the patients for whom ablation at more sites was conducted. The results presented refer to the analysis of the first 5 s signal fragment of the CS electrogram. For clarity of presentation, all measures were scaled to the (0–1) range.

**Figure 4 entropy-21-00970-f004:**
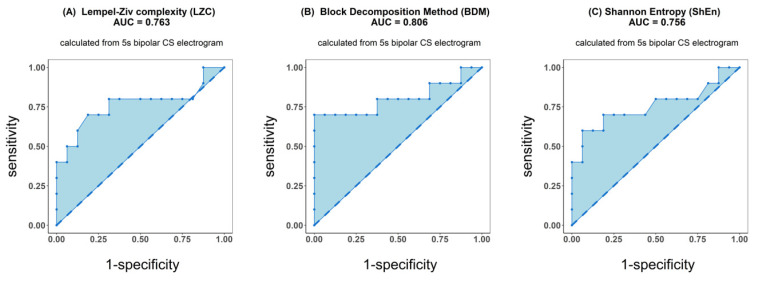
Receiver operating characteristic (ROC) analysis of classification performance for Lempel–Ziv complexity (**A**), block decomposition method (**B**) and Shannon entropy (**C**). Classification was performed between two groups of patients: those for whom AF terminated after PVI and for patients for whom ablation at more sites was conducted. Presented results refer to the analysis of the first 5 s signal fragment of the coronary sinus (CS) electrogram.

**Figure 5 entropy-21-00970-f005:**
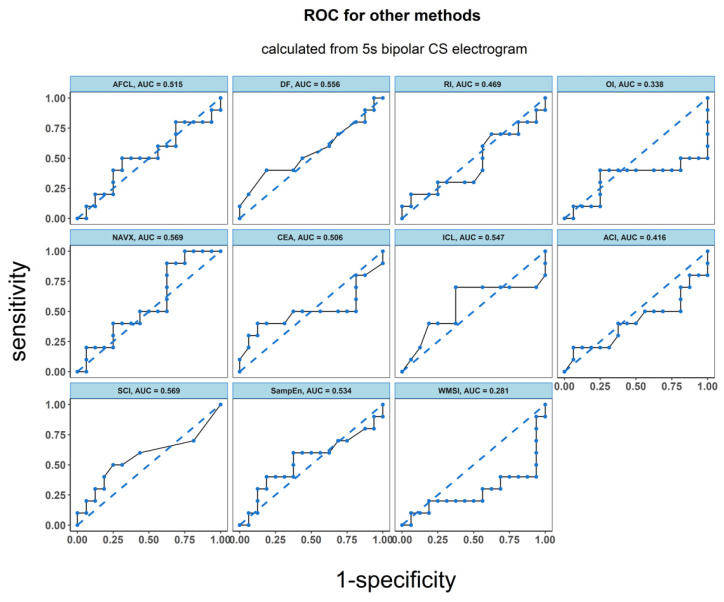
ROC analysis of the classification performance for AF cycle length, dominant frequency parameters dominant frequency (DF), regularity index (RI), organization index (OI), complex fractionated atrial electrogram (CFAE) estimated by CARTO parameters, CFAE estimated by NAVX, sample entropy (SampEn) and wave morphology similarity index (WMSI). Classification was performed between two groups of patients: the one for whom AF terminated after PVI and for those patients for whom ablation at more sites was conducted. The results presented refer to the analysis for the first 5 s signal fragment of the CS electrogram.

**Figure 6 entropy-21-00970-f006:**
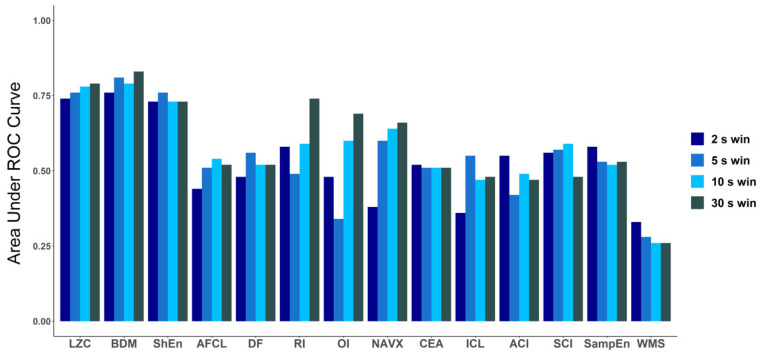
Comparison of area under ROC for all methods for different window lengths: 2 s, 5 s, 10 s and 30 s. In case of LZC, BDM and ShEn, the differences are minimal and only for those three methods in each window AUC is higher than 0.7.

**Table 1 entropy-21-00970-t001:** Results of ANOVA (*p*-values) and ROC (area under ROC) analysis for all examined methods. * For the area under ROC larger than 0.7, sensitivity, specificity and value for optimum ROC (OPT ROC) are presented. Table refers to results obtained for 5 s window analysis.

Method	*p*-Value	Area Under ROC	Sensitivity OPT ROC	Specificity OPT ROC	Value for OPT ROC
LZC	0.010	0.76	0.70	0.81	75
BDM	0.008	0.81	0.70	1.00	2076
ShanEn	0.052	0.76	0.70	0.81	2.47
AFCL	0.944	0.51	-*	-*	-*
DF	0.665	0.56	-*	-*	-*
DF RI	0.737	0.49	-*	-*	-*
DF OI	0.084	0.34	-*	-*	-*
NavX CFAE	0.876	0.60	-*	-*	-*
CEA	0.481	0.51	-*	-*	-*
ICL	0.813	0.55	-*	-*	-*
ACI	0.484	0.42	-*	-*	-*
SCI	0.293	0.57	-*	-*	-*
SampEn	0.978	0.53	-*	-*	-*
WMS	0.160	0.28	-*	-*	-*

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
