# Peer review of "Kolmogorov Complexity of Coronary Sinus Atrial Electrograms Before Ablation Predicts Termination of Atrial Fibrillation After Pulmonary Vein Isolation"

_entropy, 2019, doi:10.3390/e21100970_

Round 1

Reviewer 1 Report

Very good resubmission. I can only suggest adding a similar paper (in its methodology) recently published:

https://www.complex-systems.com/abstracts/v28_i01_a03/

It would be desirable that the authors point out another biological data research study using BDM.

Bibliography:

Citation no. 23 is incorrect, the arxiv version should not be cited when there is one version published in Entropy itself:

Entropy 20(8), 605, 2018. 

Reviewer 2 Report

This paper tests the hypothesis that the Kolmogorov complexity of a single atrial bipolar electrogram recorded during AF within the coronary sinus at the beginning of the catheter ablation may predict AF termination directly after pulmonary vein isolation (PVI). Results show significant differences in Kolmogorov complexity between patients with AF termination directly after PVI compared to patients undergoing additional ablation. Therefore, authors affirm that Kolmogorov complexity of electrograms measured at baseline before PVI can predict self-termination of AF directly after PVI. Overall, I believe that the goal of this study is very interesting and the tools used are appropriate. However, I have several major comments regarding the paper:

Authors talk about complexity measures, however they are different types of measurements and they do not measure complexity. Additional 30 second electrograms recorded 30 prior to AF termination were analyzed as the last stage and from my point of view these recordings should be analyzed to highlight the differences of different methods.

Furthermore, the authors suggest the method to be used to predict AF termination, and it is too pretentious because even they have not compared with the final recordings. In other studies, this analysis has done it, you should compare your results with previous studies. In addition, to affirm sentences such as “the  measurement of Kolmogorov complexity could be used as an indicator of the expected ablation duration and how demanding the procedure may be” or “we aimed to predict, using only electrogram properties measured at a single site, if the sole ablation of PVI will lead to acute AF termination.” you should need an additional comparison because the presented method is very weak.

Moreover, prior to the entropy calculation, certain common parameters need to be initialized: embedding dimension m, tolerance threshold r  and time series length N. Nevertheless, the determination of these parameters is usually based on expert experience. Improper assignments of these parameters tend to

bring invalid values, inconsistency and low statistical significance in entropy calculation. You should add a more detailed explanation of these parameters.

Is ventricular artifact removal necessary during AF? Is it because (some of) atrial repolarization is included? How does this affect to the measures?

There are a number of typos, e.g. line 194, 195, 526…

You should add a more detailed discussion with recent studies that have studied entropy of electrograms during AF,

Round 2

Reviewer 2 Report

Stępień et al. did a good job addressing all the comments and concerns raised by this reviewer. Moreover, you should compare your results with previous studies that apply regularity and complexity measures, but you should be very careful, your results do not suppose a long-term successful of the procedure, they only suppose that ablation would be extended beyond pulmonary veins. Then, to get more value of your results you should explain in more detail the physiology meaning of the results obtained in this study.

This manuscript is a resubmission of an earlier submission. The following is a list of the peer review reports and author responses from that submission.

Round 1

Reviewer 1 Report

This paper claims to apply a measure of algorithmic or Kolmogorov complexity but they rather apply a popular compression algorithm that has been recently proven not to be related to Kolmogorov complexity but to Shannon entropy as it was expected, hence any references to algorithmic complexity are wrong, are underserved and I would not be able to accept a paper with such claims. Now, the question is therefore if a paper using LZW as a complexity measure has any merit. In my opinion very little, as then this collapses to the use of entropy on a uniform distribution, i.e. counting symbols! in this case the density of 1s over 0s (the compression algorithm does little more, builds a dictionary, but that is a small variation of Shannon entropy to entropy rate). Therefore, I cannot support the publication of this paper. Only way to make it relevant is if the authors really take their claims seriously and, e.g. start using modern measures to approximate Kolmogorov complexity such as the so-called block decomposition method that, unlike compression algorithms, is able to pick some algorithmic signals and when it does not then switch to Shanno entropy but does not do so as the first line of defense and is justified to be called an approximation to K. Moreover, the CTM and BDM methods can deal with short strings which may be relevant to the authors. If they do so and do find similar results I will be able to support the paper.

Reviewer 2 Report

This paper shows an unclear methodology to evaluate complexity of coronary sinus atrial electrograms before ablation predicts termination of atrial fibrillation after pulmonary vein isolation. Nevertheless, the study is not precise.

From the materials section, the exact location of the bipolar electrodes in the atria is not well described. Moreover, persistent AF patients are different to paroxysmal AF patients, therefore they should be analyzed separately.

From my point of view, you mix linear and nonlinear measures, as well as complexity and regularity indexes. These issues should be reformulated. 

In addition, the authors show different methods to analyze the "complexity", moreover they do not explain the meaning and the advantages and the disadvantages of these methods. You should compare and justify their use. Moreover, 30 s length is too long recordings for the variability of AF.

Graphics should be redone because they do not provide additional information

You should rewrite the Discussion, you should add more references from previous studies that have studied complexity and regularity, as well as linear measures in electrograms during AF. You have some examples of recent studies in the following lines: 

Surface ECG and intracardiac spectral measures predict atrial fibrillation recurrence after catheter ablation.

Szilágyi J, Walters TE, Marcus GM, Vedantham V, Moss JD, Badhwar N, Lee B, Lee R, Tseng ZH, Gerstenfeld EP.

J Cardiovasc Electrophysiol. 2018 Oct;29(10):1371-1378. doi: 10.1111/jce.13699. Epub 2018 Aug 23. Select item 27448337 2.

A Multi-Variate Predictability Framework to Assess Invasive Cardiac Activity and Interactions During Atrial Fibrillation.

Alcaine A, Mase M, Cristoforetti A, Ravelli F, Nollo G, Laguna P, Martinez JP, Faes L.

IEEE Trans Biomed Eng. 2017 May;64(5):1157-1168. doi: 10.1109/TBME.2016.2592953. Epub 2016 Jul 19.

Entropy at the right atrium as a predictor of atrial fibrillation recurrence outcome after pulmonary vein ablation.

Cervigón R, Moreno J, García-Quintanilla J, Pérez-Villacastín J, Castells F.

Biomed Tech (Berl). 2016 Feb;61(1):29-36. doi: 10.1515/bmt-2014-0172.

Spatiotemporal characterization of atrialactivation in persistent human atrial fibrillation: multisite electrogram analysis and surface electrocardiographic correlations--a pilot study.Dibs SR, Ng J, Arora R, Passman RS, Kadish AH, Goldberger JJ.Heart Rhythm. 2008 May;5(5):686-93. doi: 10.1016/j.hrthm.2008.01.027. Epub 2008 Jan 29.